# Lipoprotein Lipase and Its Delivery of Fatty Acids to the Heart

**DOI:** 10.3390/biom11071016

**Published:** 2021-07-12

**Authors:** Rui Shang, Brian Rodrigues

**Affiliations:** Faculty of Pharmaceutical Sciences, The University of British Columbia, Vancouver, BC V6T 1Z3, Canada; r.shang@alumni.ubc.ca

**Keywords:** LPL, cardiac metabolism, fasting, diabetes, cardiomyopathy

## Abstract

Ninety percent of plasma fatty acids (FAs) are contained within lipoprotein-triglyceride, and lipoprotein lipase (LPL) is robustly expressed in the heart. Hence, LPL-mediated lipolysis of lipoproteins is suggested to be a key source of FAs for cardiac use. Lipoprotein clearance by LPL occurs at the apical surface of the endothelial cell lining of the coronary lumen. In the heart, the majority of LPL is produced in cardiomyocytes and subsequently is translocated to the apical luminal surface. Here, vascular LPL hydrolyzes lipoprotein-triglyceride to provide the heart with FAs for ATP generation. This article presents an overview of cardiac LPL, explains how the enzyme works, describes key molecules that regulate its activity and outlines how changes in LPL are brought about by physiological and pathological states such as fasting and diabetes, respectively.

## 1. Introduction

With uninterrupted contraction being a unique feature of the heart, the cardiomyocyte has a high demand for energy. As such, this cell demonstrates substrate promiscuity, enabling it to utilize multiple sources of energy, including fatty acids (FAs), glucose, amino acids, lactate, and ketones [1]. Among these, 95% of the ATP generated in the heart is derived from glucose and FAs, through mitochondrial metabolism. The heart cannot synthesize FAs and relies on obtaining them from other sources, including: (i) release from adipose tissue and transport to the heart; (ii) breakdown of endogenous cardiac triglyceride (TG); and (iii) lipolysis of circulating TG-rich lipoproteins to FAs by lipoprotein lipase (LPL), positioned at the endothelial cell (EC) surface of the coronary lumen [2,3]. As LPL-mediated lipolysis of lipoproteins is suggested to be a key source of FAs for cardiac use [2], its regulation and its modification following diabetes require thorough investigation to help us understand the pathophysiology of diabetic heart disease as it relates to the metabolism of FAs, in order to advance its clinical management.

## 2. Cardiac Lipoprotein Lipase—Overview

Although LPL-mediated hydrolysis of lipoproteins occurs at the EC surface in the vascular lumen, these cells do not synthesize LPL [4]. In the heart, LPL is synthesized in cardiomyocytes prior to its transfer to the vascular lumen (Figure 1). Thus, the electron microscope immunogold localization of LPL demonstrated that 78% of total LPL is present in cardiomyocytes, 3–6% in the interstitial space, and 18% at the capillary endothelium [5]. Related to its production, previous studies have suggested that LPL is synthesized as an inactive monomer in the rough endoplasmic reticulum (ER) with dimerization, and thus, enzyme activation occurs between the ER and Golgi, prior to its secretion [6,7]. A more recent study has indicated that LPL could also be active as a monomer [8].

Subsequent to its synthesis, LPL is secreted onto heparan sulphate proteoglycan (HSPG) binding sites on the cardiomyocyte apical surface [9]. At this location, positively charged LPL is attached (by ionic interaction) to negatively charged heparan sulfate (HS) side chains of HSPG. This is an effective arrangement, as this pool of LPL provides the heart with a rapidly accessible reservoir. In this way, the immediate demand for FAs can be resolved, not by attempting to synthesize more LPL, but by simply translocating the enzyme that has already been produced [10,11]. To reach the vascular lumen, LPL requires detachment from HSPG and navigation across the interstitial space, made possible by glycosylphosphatidylinositol-anchored high-density lipoprotein-binding protein 1 (GPIHBP1) [12]. GPIHBP1 is a glycoprotein abundantly expressed in the heart, exclusively in capillary ECs [13,14]. Intriguingly, on either side of these cells, GPIHBP1 accomplishes different functions. At its basolateral side, GPIHBP1 operates as a transporter, collecting LPL from the interstitial spaces surrounding myocytes and shuttling it across the ECs to the capillary lumen [15]. On the apical side of the Ecs, the ability of GPIHBP1 to avidly bind both lipoprotein-TG and LPL allows it to serve as a platform for the lipolytic processing of TG, which releases FAs for use by cardiomyocytes [15,16,17]. It should be noted that, related to the function of LPL, its action on chylomicron-TG clearance is likely more significant than its effect on very low-density lipoprotein (VLDL)-TG hydrolysis. Chylomicrons are larger in size, have a lower density and a greater amount of TG. This increases the chance of this lipoprotein particle interacting with LPL at the coronary vascular lumen [18]. An additional difference between the two lipoproteins and LPL action is that lipolysis of VLDL-TG produces FAs that are consequently transported into cells via CD36, an FAs transporter, whereas hydrolysis of chylomicrons generates a greater local concentration of FAs that enter cells by passive flip-flop and/or non-CD36-mediated transport mechanisms [19].

## 3. Posttranslational Processes That Regulate Cardiac LPL

The cardiac muscle fulfills its energy demand by preferentially using FAs. To obtain this substrate, the heart would prefer mechanisms to control its own delivery of FAs, rather than depending on FAs from other organs, similar to adipose tissue. In this regard, the heart makes use of LPL-derived FAs to provide itself with a substantial amount of energy. Multiple regulators exist that oversee cardiac LPL action (Figure 1), and include:

### 3.1. AMP-Activate Protein Kinase

AMP-activated protein kinase (AMPK) is a cellular energy sensor that is activated upon energy deprivation. Upon activation, AMPK modulates downstream effectors via its kinase activity to augment ATP production. One mechanism by which this is achieved is through its inhibition of acetyl-CoA carboxylase (ACC) [20]. As ACC is responsible for the generation of malonyl-CoA, a key intermediate metabolite that blocks carnitine palmitoyltransferase-1 (CPT-1), and thus the transport of FAs into the mitochondria, AMPK activation facilitates the oxidation of FAs [1]. AMPK activation has also been implicated in the delivery of FAs to cardiomyocytes through its regulation of the FAs transporter CD36 [21]. Results from our laboratory have demonstrated a strong correlation between the phosphorylation of AMPK and increases in coronary lumen LPL activity [22]. Specifically, by phosphorylating heat shock protein 25 (Hsp25), AMPK causes the dissociation of Hsp25 from actin monomers, leading to actin cytoskeleton polymerization and the transport of LPL-containing vesicles from the Golgi to the plasma membrane. In this way, intracellular LPL is moved to the cardiomyocyte surface and eventually the vascular lumen [3]. Given the role of AMPK as a key signaling molecule, conditions that modulate AMPK are associated with switching on coronary LPL activity. For example, during fasting with limited carbohydrate availability, the activation of AMPK, and thus augmented LPL, is an adaptation that would ensure adequate cardiac energy [23]. AMPK is also activated in response to acute hypoinsulinemia in moderately diabetic rats, possibly because impaired glucose utilization causes energy deficiency. In these animals, this resulted in higher coronary LPL activity, which was responsible for changing the substrate focus of the heart to predominantly using FAs [9]. It should be noted that severe diabetes introduces an additional metabolic manifestation in these animals, hyperlipidemia [24], that is known to inhibit AMPK in the heart [25]. This is followed by a corresponding reduction in coronary LPL activity [26]. The above evidence indicates that AMPK activation contributes towards LPL trafficking, and thus the cardiac utilization of FAs.

### 3.2. Heparanase

Heparanase is an endoglycosidase that is synthesized in ECs as an inactive, latent 65 kDa enzyme (L-Hpa) that undergoes cellular secretion followed by HSPG-facilitated reuptake [27,28]. After proteolytic cleavage in lysosomes, a 50 kDa polypeptide is formed that is at least 100-fold more active (A-Hpa) than L-Hpa [29,30]. Lysosomes store A-Hpa until they are mobilized by different stimuli. In normal physiology, heparanase has a function in embryonic morphogenesis, wound healing and hair growth [31]. Our lab was the first to describe a unique responsibility of heparanase in cardiac metabolism; releasing myocyte LPL for forward movement to the vascular lumen to provide the heart with FAs [32]. In conditions of pathology, such as Type 2 diabetes, high levels of heparanase have been reported in plasma and urine [33,34]. Using perfused hearts, we established that acute hyperglycemia had a robust influence on heparanase secretion into the interstitial space [35]. We also discovered that incubating ECs in high glucose conditions triggered lysosomal A-Hpa to be released into the medium [36]. In high glucose conditions, there is a redistribution of lysosomal heparanase from a perinuclear location towards the plasma membrane of ECs. ATP release, purinergic receptor activation, cortical actin disassembly and stress actin formation were essential for high glucose-induced heparanase secretion [35]. These studies fixated on the effects of high glucose on A-Hpa secretion, as we incorrectly assumed that only the HS hydrolyzing ability of heparanase would be successful in releasing myocyte surface-bound proteins. With additional reasoning, we realized that it would be futile for a cell to secrete an inactive protein for subsequent reuptake and activation unless the enzymatically inactive L-Hpa also serves a function. Intriguingly, L-Hpa has some remarkable properties, including its ability to mediate HSPG clustering to activate p38 mitogen-activated protein kinase, Src, PI3K-Akt, and RhoA [32,37,38,39,40]. This transmembrane signaling could then reload the cardiomyocyte surface pool of LPL that was liberated by A-Hpa. Overall, heparanase plays an important role in EC–cardiomyocyte crosstalk to affect LPL translocation, and thus the delivery of FAs to the heart. 

### 3.3. GPIHBP1

GPIHBP1 is expressed exclusively in capillary ECs, while LPL is expressed in parenchymal cells such as cardiomyocytes [41]. Given its previously described role in LPL translocation and function, it is not surprising that GPIHBP1-deficient mice developed severe hyperlipidemia even when fed a low-fat chow diet [15]. Similarly, patients with GPIHBP1 mutations have severe hypertriglyceridemia [42]. Related to its structure, GPIHBP1 has a GPI anchor, and thus can be released from the plasma membrane by cleaving off this anchor with phosphatidylinositol-specific phospholipase C [13]. It is the *N*-terminal acidic domain of GPIHBP1 that can electrostatically interact with LPL [43]. In addition to GPIHBP1′s ability to facilitate EC LPL transport and provide a platform for lipoprotein hydrolysis, an added function of GPIHBP1 has emerged, whereby it reduces the unfolding of the catalytic domain of LPL by angiopoietin-like protein 4 (ANGPTL4), consequently stabilizing LPL activity [44,45,46]. Evidence that GPIHBP1 expression is regulated came from experiments investigating the effects of fasting/refeeding [47]. Fasting amplifies cardiac GPIHBP1 expression, an effect that is reversed 6 h after refeeding [47]. In hearts from animals with diabetes, GPIHBP1 gene and protein expression increased with exclusive EC localization [48]. Moreover, the exposure of ECs to high glucose conditions also uncovered a rapid increase in GPIHBP1 mRNA and protein [41,49]. Interestingly, ECs exposed to recombinant L-Hpa showed a substantial time-dependent increase in the GPIHBP1 gene and protein [41]. As this effect was also duplicated by active heparanase, high glucose, together with its secretion of heparanase, could be the unforeseen mechanism by which the EC can increase its expression of GPIHBP1 to augment the supply of FAs following diabetes.

### 3.4. Fatty Acid

Multiple mechanisms have been suggested to explain the role of FAs in regulating LPL. These include: (a) the impairment, by FAs, of LPL vesicle trafficking to the myocyte surface through the caspase-3-mediated cleavage of protein kinase D [50]; (b) suppression, by FAs, of EC heparanase secretion, thereby reducing the translocation of cardiomyocyte LPL [35]; (c) displacement of luminal LPL for degradation in the liver by FAs [51]; and (d) the direct inactivation of LPL by FAs [52]. The idea that LPL can also be inactivated by the induction of ANGPTL4 by FAs has also been suggested [53,54,55]. This idea arose from a study where ANGPTL4 deficiency caused an elevation of post-heparin plasma LPL activity in mice [56]. ANGPTL4 is widely expressed in various tissues, including white adipose tissue, liver, heart, and skeletal muscle [54]. In cells from these tissues, it forms oligomers through disulfide linkage, and, after secretion, is cleaved at a canonical proprotein convertase cleavage site that separates the *N*- and *C*-terminal domain [57]. The *N*-terminal coiled-coil domain remains oligomerized, a feature that is essential for its stability and an LPL inhibitory effect [58]. Historically, the secreted *N*-terminal oligomers were thought to act in the circulation, converting dimeric LPL at the vascular lumen into inactive monomers [53]. More recently, a study has suggested that ANGPTL4 functions to unfold active LPL monomers, thus reducing LPL activity [59]. This inhibitory function of ANGPTL4 was not evident for GPIHBP1-stabilized LPL [45,60]. Circulating FAs can increase ANGPTL4 gene transcription [61,62] and we have also reported that cardiomyocytes exposed to FAs demonstrate higher ANGPTL4 expression [26]. Related to pathology, severely diabetic animals with hyperlipidemia showed a decline in heparin-releasable LPL activity at the vascular lumen [26]. This effect was not related to any reduction in LPL gene expression [63] and implied that the low LPL activity is likely an upshot of post-translational mechanisms. Strikingly, transcriptome profiling by RNA-seq revealed that of the 1574 differentially expressed genes in these diabetic hearts, the ANGPTL4 gene was the one with the highest fold change (~25-fold increase) [24]. At present, the location where ANGTPL4 inactivates cardiac LPL is yet to be determined. Additionally, similar to ANGPTL4, ANGPTL3 (in a complex with ANGPTL8) is mainly produced by the liver, and is also known to influence LPL activity [64]. Collectively, these studies provide us with clues to how FAs “turn off” vascular LPL, through multiple mechanisms, to avoid lipid oversupply to the heart. 

### 3.5. Insulin

The unique characteristic of LPL is that it is rapidly responsive to variations in circulating insulin and does so in a tissue-specific manner. Thus, in tissues such as adipose tissue and the heart, changes in LPL activity can occur independent of alterations in mRNA [65]. Such changes would be desirable given that, during conditions such as fasting or diabetes, wherein insulin levels are altered and the utilization of FAs is augmented, a rapid requirement for LPL may not be matched by the slow turnover of LPL mRNA. Related to tissue specificity, the lowering of circulating insulin following fasting decreases LPL activity in adipose tissue but increases it in the heart [66]. These changes in luminal LPL activity following fasting were independent of shifts in LPL mRNA or alterations in LPL protein, suggesting a posttranslational mechanism for this increase [23]. As a result, FAs generated from circulating TG are diverted away from storage, towards meeting the metabolic demands of cardiomyocytes (it fulfills a “gate-keeping” role by regulating the supply of FAs to meet tissue requirements). As in perfused guinea pig hearts, newly synthesized LPL can move from myocytes to the vascular lumen within 30 min. An increased rate of LPL transfer could explain this effect of fasting on coronary LPL [67]. Mechanistically, because glucose entry into the cardiomyocyte is largely dependent on insulin action, a fasting- or diabetes-induced reduction in insulin causes a rise in the intracellular AMP-to-ATP ratio, with a resultant activation of AMPK [2,68,69]. Once stimulated, AMPK switches off energy-consuming processes such as protein synthesis, whereas ATP-generating mechanisms, such as the oxidation of FAs and LPL translocation, are turned on [3].

## 4. The Consequences of Oscillations in LPL

As indicated, the nutritional regulation of cardiac LPL determines the delivery of FAs to the heart in order to maintain energy demand [66]. However, oscillations in LPL have also been linked to a disruption in cardiac metabolism, leading to pathology.

### 4.1. Gain-and Loss-of-Function of Cardiac LPL

The surplus provision of FAs to tissues other than the adipose tissue can trigger cellular demise [3,70,71,72]. Not surprisingly, then, muscle-specific LPL overexpression causes severe myopathy, characterized by lipid oversupply, the proliferation of mitochondria and peroxisomes, muscle fiber degeneration, excessive dilatation and impaired ventricular function in the absence of vascular defects [73,74]. LPL is also known to play a role in insulin signaling (Figure 1). Hence, transgenic mice with muscle-specific LPL overexpression exhibited skeletal muscle and whole-body insulin resistance [75], and variations in the LPL gene have been reported to play a role in determining insulin resistance in Mexican Americans [76]. In contrast, the loss of cardiac LPL also causes heart failure [77,78]. Interestingly, although this increases glucose use, neither this effect, nor albumin-bound FAs, could replace the action of LPL, and the cardiac ejection fraction decreased [78]. These experiments in genetically modified models suggest that disturbing cardiac LPL is sufficient to induce cardiac failure. 

### 4.2. Fluctuations in Cardiac LPL Following Diabetes

In patients, heparin is used to displace LPL from its binding sites, enabling the subsequent determination of its plasma activity [11,79]. The drawback with this approach is that it measures LPL released from all tissues (skeletal muscle, adipose tissue, heart), and hence is incapable of establishing if diabetes specifically influences cardiac LPL. Regarding tissue-specific assessment, adipose tissue and skeletal muscle show low levels of LPL in homogenates following diabetes [80], with virtually no information available on the cardiac content of this enzyme. Even if tissue specificity can be resolved by using homogenates, the estimation of cardiac LPL from patients with diabetes would be inappropriate, as it would reflect total cardiac LPL and not the more pertinent functional pool at the coronary lumen. Hence, data from animal studies have provided the majority of information regarding cardiac LPL in diabetes. 

Following acute insulin resistance [81,82] or moderate hypoinsulinemia and hyperglycemia in rats [9,26,83,84], LPL is “switched on”, causing a robust expansion of heparin-releasable coronary LPL. We defined a model where only a fraction of the binding sites at the luminal side of coronary EC are occupied by LPL, and hyperglycemia leads to rapid filling of the unoccupied sites [65]. The increase in LPL at this location was immediate [65,83,85] and unrelated to gene and protein expression [83,86]. Instead, it occurred through post-translational mechanisms that increase the transfer of LPL from the underlying cardiomyocyte to the apical side of the ECs [87]. Transfer to the coronary lumen requires the movement of LPL to the cardiomyocyte plasma membrane [65], controlled by AMPK [23,88], p38 mitogen-activated protein kinase and protein kinase D [50,63,89]. The activation of these kinases facilitates LPL vesicle formation and cytoskeletal rearrangement for secretion onto cardiomyocyte cell-surface HSPG [63,90]. For its onward movement, the detachment of LPL from the cardiomyocyte surface is a prerequisite and is mediated by the cleavage of HSPG by heparanase [32,35]. In response to high glucose conditions, ECs secrete heparanase [35,36], thereby instigating cardiomyocyte LPL release [91]. In addition to LPL, heparanase also functions to release cardiomyocyte vascular endothelial growth factor A (VEGFA) [41,92] and VEGFB [93], prospective partners that, by modulating both O_2_ delivery and inhibiting cell death, can offer safeguards when FAs are being used disproportionately. Besides severe diabetes [9,26], a decline in LPL has also been observed in animals infused with Intralipid [94]. As these animals exhibit elevated plasma FAs, we concluded that the LPL-mediated delivery of FAs would be redundant in these circumstances, and is “turned off”.

### 4.3. Lipid Metabolites and Diabetic Cardiomyopathy

In patients and animals with Type 1 and Type 2 diabetes, there is a reduced or low–normal diastolic function and left-ventricular hypertrophy in the absence of vascular defects, emblematic of diabetic cardiomyopathy [95,96,97,98,99,100]. Several factors have been associated with the development of diabetic cardiomyopathy and include an accumulation of connective tissue and insoluble collagen, impaired sensitivity to various ligands, mitochondrial dysfunction, ER stress, RAAS activation and abnormalities in proteins that regulate intracellular Ca^2+^ [98,101,102,103]. However, with diabetes and augmentation of the availability of FAs, either through an increase in vascular LPL or adipose tissue lipolysis, the diabetic heart has a mismatch between delivery of FAs and their oxidation, leading to myocyte lipid droplet synthesis and the accumulation of lipid metabolites (e.g., DAG, LPC and ceramide). We and others have proposed that this mediates lipid-induced insulin resistance, cell death and eventually diabetic cardiomyopathy [70,71,72,104,105]. As the chronic management of glycemia fluctuates over the duration of diabetes, with frequent episodes of insufficient or poor control, pathological oscillations in coronary LPL arise during this disease to disturb cardiac metabolism and initiate heart failure.

## 5. Conclusions

Following diabetes, the heart shifts to the predominant use of FAs, leading to the development of diabetic cardiomyopathy. Accordingly, treatment of this cardiac syndrome will require us to rethink our therapeutic strategies, from a focus on controlling blood glucose to re-establishing physiological cardiac metabolism. Understanding metabolic disequilibrium and lipotoxicity will allow the identification of targets for a mechanism-driven therapeutic intervention to help prevent/delay cardiovascular disease, characteristic of diabetes [106].

## Figures and Tables

**Figure 1 biomolecules-11-01016-f001:**
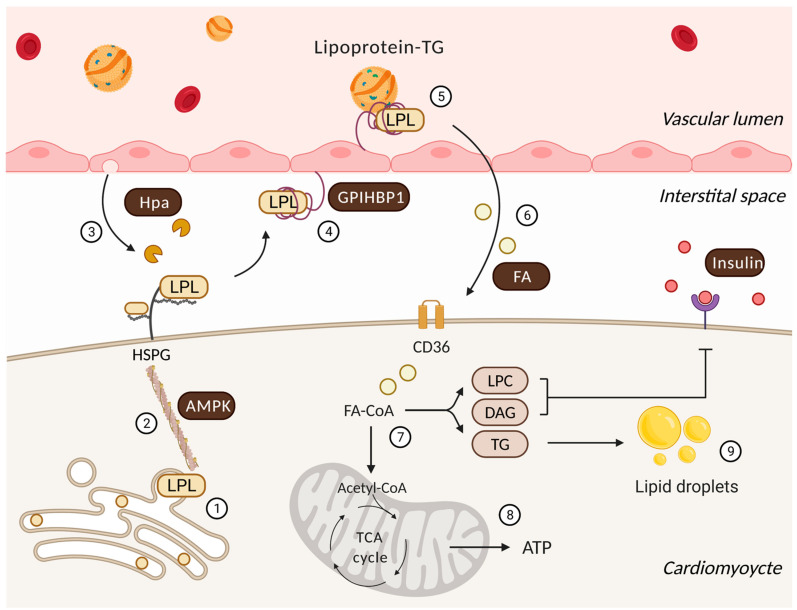
Regulatory processes that influence cardiac lipoprotein lipase action and the utilization of delivered FAs. Following its synthesis and activation in cardiomyocytes (**1**), the actin cytoskeleton traffics LPL to myocyte cell surface HSPG under the control of AMPK (**2**). Onward progress is facilitated by heparanase (Hpa) secreted from the endothelial cell that cleaves HSPG side chains to liberate LPL (**3**). GPIHBP1 captures this interstitial LPL and relocates it from the basolateral to the apical side of endothelial cell (**4**). At this location, a GPIHBP1 platform and LPL enables lipoprotein-TG hydrolysis to generate FAs (**5**) that are delivered to the cardiomyocyte (**6**). In the cardiomyocyte, FAs have two major fates (**7**). They can either undergo mitochondrial beta-oxidation and oxidative phosphorylation to generate ATP (**8**) or accumulate as lipid metabolites/droplets (**9**). Lipid intermediates are known to effect insulin signaling and substrate utilization. LPC: lysophosphatidylcholine; DAG: diacylglycerol.

## Data Availability

Not applicable.

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
