# Peer review of "Lipoprotein Lipase and Its Delivery of Fatty Acids to the Heart"

_biomolecules, 2021, doi:10.3390/biom11071016_

Round 1
Reviewer 1 Report
Review Report
- A brief summary
The present review titled “Lipoprotein lipase and its delivery of fatty acid to the heart” presents an overview of cardiac LPL, explains how the enzyme works, describes key molecules that regulate its activity and outlines how changes in LPL are brought about by physiological and pathological states like fasting and diabetes respectively.
Special attention is given to the mechanism of LPL-mediated lipolysis of lipoproteins as a key source of FA for cardiac energy, as well as its regulation and modification following diabetes. In the review is consistently considered the synthesis and secretion of cardiac lipoprotein lipase, posttranslational processes that regulate cardiac LPL which includes AMP-activate protein kinase which modulates downstream effectors leading to augmentation of ATP production, the responsibility of heparanase in cardiac metabolism, the role of GPIHBP1 in LPL translocation and function, fatty acid and insulin as activators and regulators of cardiac LPL.
The review also pays attention to the pathology raised by oscillations in LPL. That includes Gain-and loss-of-function of cardiac LPL, fluctuations in cardiac LPL following diabetes, lipid metabolites, and diabetic cardiomyopathy.
In summary, the present review could give the opportunity for better therapeutic intervention to prevent or delay cardiovascular disease following diabetes by understanding metabolic disequilibrium and lipotoxicity.
Novelty, the significance of the content and interest to the readers
The review provides many promising aspects for rethinking therapeutic strategies for the treatment of cardiac syndrome after diabetes in order to restore the physiological cardiac metabolism, which is a prerequisite for arousing great interest in readers.
Quality of presentation and scientific soundness
The quality of the presentation is on a very high level of organization. This is helped by the good structuring of the review and the added schemes, which contribute to the easy assimilation of the provided information. The correct-chosen references also contribute to the excellent performance of the review, most of which are from the last decade. The paper is properly designed with a high level of expertise.
Specific comments
In many places in the text, the full names of the abbreviations are omitted where they first appear. Please add them.

Author Response
We thank the reviewer for these positive comments. Related to your specific comment about abbreviations, we have closely proofread the document and have made the changes accordingly.
Reviewer 2 Report
The authors reviewed lipoprotein lipase and its delivery of fatty acid to the heart. They included one figure and 104 references. They provided an overview of cardiac lipoprotein lipase, its posttranslational processes and specially effect of diabetes mellitus. The manuscript is well written and easy to follow. They provided up-to-date information and a lot of readers will enjoy it.
Author Response
We thank the reviewer for this positive feedback.
Reviewer 3 Report
Lipoprotein lipase is the key enzyme for the hydrolysis of plasma triglycerides and due to a diverse regulation of its activity on different levels, the body is able to meet the different requirements of the target organs such as adipose tissue or muscles during the fasting or postprandial status.
In their wonderful and comprehensive review article, Rui Shang and Brian Rodrigues summarize the importance of lipoprotein lipase for supplying the heart with free fatty acids. In their introduction, they skilfully circumvented the current dilemma in which form LPL is active: as a dimer, as has been the textbook opinion for decades, or as a monomer? Then they go into detail on different mechanisms of how the regulation of the LPL is organised specifically in the heart. In their very extensive list, however, I miss a description of the function of ANGPTL3 / 8. In this context, it would be nice if they include the ANGPTL proteins and their proposed place of action in Figure 1.
Author Response
Thank you for that suggestion. We have now added the following to the revise manuscript.
"At present, the location where ANGTPL4 inactivates cardiac LPL is yet to be determined. Additionally, like ANGPTL4, ANGPTL3 (in a complex with ANGPTL8) mainly produced by the liver, are also known to influence LPL activity [64]."
.